# Paravalvular Aortic Regurgitation Severity Assessed by Quantitative Aortography: ACURATE *neo*2 versus ACURATE *neo* Transcatheter Aortic Valve Implantation

**DOI:** 10.3390/jcm10204627

**Published:** 2021-10-09

**Authors:** Andreas Rück, Won-Keun Kim, Hideyuki Kawashima, Mahmoud Abdelshafy, Ahmed Elkoumy, Hesham Elzomor, Rutao Wang, Christopher U. Meduri, Dinos Verouhis, Nawzad Saleh, Yoshinobu Onuma, Darren Mylotte, Patrick W. Serruys, Osama Soliman

**Affiliations:** 1Department of Cardiology, Karolinska University Hospital, SE-171 76 Stockholm, Sweden; andreas.ruck@sll.se (A.R.); chrismeduri@gmail.com (C.U.M.); dinos.verouhis@sll.se (D.V.); Nawzad.saleh@sll.se (N.S.); 2Kerckhoff Heart Center, Department of Cardiology, 61231 Bad Nauheim, Germany; W.Kim@kerckhoff-klinik.de; 3Discipline of Cardiology, Saolta Group, Galway University Hospital, Health Service Executive and CORRIB Core Lab, National University of Ireland Galway (NUIG), H91 V4AY Galway, Ireland; h.kawashima429@gmail.com (H.K.); mahmoud.abdelshafy@nuigalway.ie (M.A.); Ahmed.Elkoumy@nuigalway.ie (A.E.); hesham.elzomor@nuigalway.ie (H.E.); rutao.wang@nuigalway.ie (R.W.); yoshinobu.onuma@nuigalway.ie (Y.O.); darrenmylotte@gmail.com (D.M.); patrick.w.j.c.serruys@gmail.com (P.W.S.); 4Academic Medical Centre, Department of Cardiology, University of Amsterdam, P.O. Box 22660, 1100 DD Amsterdam, The Netherlands; 5Department of Cardiology, Radboud University Medical Center, 6500 HB Nijmegen, The Netherlands; 6CÚRAM, the SFI Research Centre for Medical Devices, H91 TK33 Galway, Ireland; 7NHLI, Imperial College London, London SW7 2AZ, UK

**Keywords:** transcatheter aortic valve implantation, transcatheter heart valve, aortic regurgitation, ACURATE *neo*, videodensitometry

## Abstract

The new-generation ACURATE *neo*2 system was commercially released in September 2020. In this study, we sought to compare the aortic regurgitation (AR) severity of the ACURATE *neo*2 versus the ACURATE *neo* transcatheter heart valve, using quantitative videodensitometric angiography (qAR). This is a retrospective, Corelab analysis of final post-transcatheter aortic valve implantation (TAVI) aortograms of patients treated with the ACURATE *neo*2 and ACURATE *neo* systems. The ACURATE *neo*2 cohort comprised consecutive patients treated between September 2020 and January 2021 at two centers. The ACURATE *neo* cohort included consecutive patients treated before September 2020. Our primary objective was to compare AR severity on qAR following TAVI with ACURATE *neo*2 and ACURATE *neo*. Out of 401 aortograms, 228 (56.9%) were analyzable, with 120 in the ACURATE *neo*2 cohort, and 108 in the ACURATE *neo* cohort. The mean AR fraction was 4.4 ± 4.8% in the *neo*2 cohort, and 9.9 ± 8.2% in the *neo* cohort (*p* < 0.001). Furthermore, moderate or severe AR (qAR > 17%) was detected in 2 aortograms (1.7%) in the *neo*2 cohort and 15 aortograms (13.9%) in the *neo* cohort (*p* < 0.001). Quantitative aortography shows a lower rate of moderate or severe paravalvular AR in what is the first European experience of the new-generation, self-expanding ACURATE *neo*2 when compared to the first-generation ACURATE *neo*. Moreover, aortographic data need to be correlated and compared to Core Laboratory-adjudicated 30-day echocardiographic data.

## 1. Introduction

Moderate or severe aortic regurgitation (AR) following transcatheter aortic valve implantation (TAVI) has been associated with increased short- and long-term mortality [1]. In the randomized SCOPE-2 trial [2] comparing the ACURATE *neo* transcatheter heart valve (THV) (Boston Scientific Corporation, Natick, Massachusetts, USA) with the Evolut THV series (Medtronic, Minneapolis, MN, USA), the rates of cardiac death were 2.8% vs. 0.8% (*p* = 0.03) at 30 days and 8.4% vs. 3.9% (*p* = 0.01) at one year, respectively. Excess mortality was partially attributed to the higher, 30-day rate of moderate or severe paravalvular AR in the ACURATE *neo* arm (10% vs. 3%; *p* = 0.002). The newly designed, self-expanding ACURATE *neo*2 THV (Boston Scientific Corporation, Natick, MA, USA) is equipped with inner and outer pericardial skirts extended to cover the waist of the stent in order to improve conformability to calcified and irregular aortic valve anatomy, thereby preventing or mitigating paravalvular AR.

Quantitative videodensitometric angiographic assessment of aortic regurgitation (qAR) relies on time–density curves recorded in the region of reference (aortic root) and in the region of interest (left ventricular outflow tract (LVOT)) [3,4,5]. The qAR has been extensively vetted and validated in vitro [6,7], in animal models [8], and in a clinical setting in comparison to transthoracic and transoesophageal echocardiography [9,10], as well as cardiac magnetic resonance imaging [11]. Furthermore, the long-term vital prognostic value of a threshold of 17% in AR has been reported [12]. The improvement in AR following post-balloon dilatation has also been assessed with this technique, and its impact on long-term prognosis has been demonstrated [13].

In the present study, we aim to compare the severity of paravalvular AR, as assessed by qAR, in two cohorts of patients treated either with the new-generation ACURATE *neo*2 THV or the first-generation ACURATE *neo* THV.

## 2. Materials and Methods

This is a retrospective analysis of the final post-TAVI aortogram from patients treated with TAVI using the ACURATE *neo*2 and ACURATE *neo* THVs in a Core Laboratory, independent of industry. The ACURATE *neo*2 cohort comprised consecutive patients treated between September 2020 and January 2021 at two centers (Karolinska University hospital, Stockholm, Sweden and Kerckhoff Heart Center, Bad Nauheim, Germany), and participating in the multicenter Early *Neo*2 Registry (NCT04810195). Likewise, the ACURATE *neo* cohort included consecutive patients treated before September 2020. The consecutive recruitment of patients was a prerequisite for this analysis. Patients with severe aortic stenosis (AS) were treated with TAVI, and this was based on the decision of the local heart team. The study protocol was developed in accordance with the Declaration of Helsinki, and was approved by the ethics committee of each participating hospital. Data acquisition and analysis were performed in compliance with protocols approved by the Ethical Committee of the Karolinska University (NCT04810195). Written informed consent was obtained from all participants prior to the study.

A quantitative angiographic videodensitometric assessment of paravalvular AR was performed using the CAAS A-Valve 2.0.2 (Pie Medical Imaging BV, Maastricht, The Netherlands). Details of the Core Laboratory methodology are described elsewhere [9,10,11,12,13,14,15,16,17,18]. Aortographic data were analyzed in an independent Core Laboratory (CORRIB Research Center for Advanced Imaging and Core Lab, Galway, Ireland) by experienced analysts who were blinded to the investigators and to other clinical data. When analyzing the angiographies, the difference between ACURATE *neo*2 and ACURATE *neo* THVs was not detectable.

ACURATE *neo*2 and ACURATE *neo* THV sizing was conducted according to manufacturer instructions, and based on the preprocedural multidetector computed tomographic and echocardiographic measurements. A perimeter-derived mean annulus diameter was used for size selection. Computed tomography (CT) acquisition and analysis were performed according to the local practice of each participating site. TAVI procedures were performed via the transfemoral approach in all patients. Used THVs included the ACURATE *neo*2 (23, 25, and 27 mm) and the ACURATE *neo* (23, 25, and 27 mm).

The main outcome of the study was understanding the severity of paravalvular AR, assessed by qAR following TAVI. Both the absolute value of AR fraction (between 0 and 100%) as well as grade of severity (none or trace; mild; moderate or severe) were used to compare THV performance between the ACURATE *neo*2 and the ACURATE *neo* THVs. The stratification of continuous variable regurgitation fractioninto categorical variables was performed according to the following predetermined threshold criteria: (1) none or trace regurgitation (qAR < 6%); (2) mild (6% ≤ qAR ≤ 17%); and (3) moderate or severe (qAR > 17%) [9,10,11,12,13,14,15,16,17,18]. No other outcome variables were assessed in this study.

Categorical variables were reported as numeric values and percentages, and compared with the chi-square test or Fisher’s exact test as appropriate. The mean ± standard deviation for continuous variables was compared using the Student *t-*test or the Mann–Whitney U-test, depending on the variable distribution. We compared baseline and procedural characteristics for potential selection bias between the ACURATE *neo*2 and ACURATE *neo* cohorts. The proportion of patients with moderate or severe AR (qAR > 17%) following TAVI was compared using the chi-square test. A two-sided *p* value of 0.05 was considered indicative of statistical significance. Statistical analyses were performed with SPSS version 26.0 (IBM, Armonk, New York, NY, USA).

## 3. Results

Among the 401 patients included in this study, no final aortogram injection was performed in 25 (6.2%) patients, while qAR was analyzable in 228 (60.6%) patients, including 120 and 108 patients treated with the ACURATE *neo*2 and ACURATE *neo* THVs, respectively. The common causes of the non-analyzability of post-TAVI aortograms are listed in Figure 1. Out of 148 non-analyzable cases, the main reasons provided were the overlapping of the descending aorta with LVOT (48.0%) and the overlapping of the descending aorta on ascending aorta (28.4%) (Figure 1). The mean age and the Euro score II were not significantly different between the ACURATE *neo*2 and ACURATE *neo* cohorts (80.9 ± 6.1 vs. 80.4 ± 6.2, *p* = 0.485, and 4.6 ± 3.7 vs. 5.5 ± 6.7, *p* = 0.237). Baseline characteristics, including cardiovascular risk factors, comorbidities, and hemodynamic parameters on echocardiography, were similar between patients treated with the ACURATE *neo*2 and ACURATE *neo* THVs (Table 1). Likewise, procedural characteristics were similar between the two cohorts, with the exception that predilatation was used less frequently (70.0% vs. 100%, *p* < 0.001) in the ACURATE *neo*2 cohort (Table 1). Post-procedure, there were no significant differences in complications that followed TAVI between the two cohorts (Table 1).

The mean post-TAVI aortic regurgitation fraction was lower in the ACURATE *neo2* when compared with the ACURATE *neo* (4.4 ± 4.8% vs. 9.9 ± 8.2%; *p* < 0.001) (Figure 2). In addition, the rate of moderate or severe AR was lower for the ACURATE *neo*2 than for the ACURATE *neo* (1.7% vs. 13.9%, *p* < 0.001) (Figure 3 and Figure 4).

## 4. Discussion

To the best of our knowledge, this is the first study to compare the post-TAVI paravalvular AR of the self-expanding ACURATE *neo*2 THV with the first-generation ACURATE *neo* THV. The quantitative aortographic analysis reveals a 12.2% absolute risk reduction in the rate of moderate or severe AR with the ACURATE *neo*2 when compared with the ACURATE *neo* THV.

The new-generation ACURATE *neo*2 THV system was commercially released in September 2020 to replace the first-generation ACURATE *neo* THV. The newly designed valve system features the same self-expanding nitinol frame, porcine pericardial leaflets, and delivery system as the earlier generation ACURATE *neo* THV, with the exception of a modified skirt material and coverage [19]. The newly designed ACURATE *neo2* is equipped with a 60% larger inner and outer skirt that covers the inflow and the waist of the stent. Furthermore, the redesigned skirt is made of a specific material to comply with the calcified and irregular annulus anatomy in the device landing zone. The ACURATE *neo*2 is also equipped with radiopaque positioning markers for accurate placement, which might have aided in mitigating the severity of paravalvular AR of the valve. Our analysis demonstrated that the ACURATE *neo*2 THV is associated with a significant reduction in the aortic regurgitation fraction and a lower rate of moderate or severe paravalvular AR, in comparison with the ACURATE *neo* THV. This can be explained by how the internal skirt of the ACURATE *neo*2 THV prevents the bioprosthetic valve from inadvertent damage caused by native calcium spicules, and thus minimizes propensity for AR. Additionally, as mentioned previously, the extended frame coverage of the ACURATE *neo*2 by the external skirt mitigates paravalvular AR by facilitating the plugging of micro-channels at the THV anchor site.

We used qAR, a quantitative videodensitometric aortography software, in this comparison study. In the prospective RESPOND study, the qAR displayed a good relationship with the Core Laboratory-adjudicated echocardiographic, providing a more granular discrimination of regurgitation within the same strata of regurgitation as assessed by echocardiography [15]. Furthermore, this qAR is used as part of the primary composite end-point in the study protocol of the randomized LANDMARK trial (NCT04275726), comparing the Myval THV with the Evolut and Sapien 3 THV series [20].

This study included consecutive patients treated with TAVI at two European centers using the ACURATE *neo* THV systems. Essentially, the first series of patients treated with the ACURATE *neo*2 THV, representing the index cohort, were compared to the latest series of patients treated with the ACURATE *neo* THV. The change from the earlier-generation ACURATE *neo* to the newly designed THV occurred in September 2020. Baseline characteristics, including the aortic annulus perimeter on CT scan, were similar between the two cohorts. In addition, all procedures were performed with the same highly experienced operators in performing TAVI using the ACURATE *neo* THV systems. The only significant difference between the two cohorts was the reduced use of predilatation in the ACURATE *neo*2 when compared with the ACURATE *neo*. It is unlikely that the infrequent use of predilatation in the ACURATE *neo*2 cohort played a role in the reduction in AR severity. However, the association between predilatation and the post-TAVI AR severity has yet to be investigated.

Our findings, although meticulously analyzed by highly experienced observers in a Core Laboratory setting, should be considered as hypothesis-generating, and thereby should be interpreted in line with the following study limitations. Firstly, these data are derived from two large-volume European centers, and by TAVI operators highly experienced in using the ACURATE THV systems. Therefore, the generalizability of our findings of improved performance of the ACURATE *neo* 2 THV system needs further confirmation in a larger population, including more operators and more centers. In addition, this study was focused on comparing the acute AR severity between the two cohorts, and no post-TAVI echocardiographic data were reported. Therefore, the next logical step is to correlate and compare aortographic data to Core Laboratory-adjudicated 30-day echocardiographic data. Finally, the durability of the ACURATE *neo*2 THV was not investigated. Further studies comprising at least one year of clinical and echocardiographic follow-up, including an independent clinical event committee and Core Laboratory adjudications, are needed to ascertain our preliminary findings on the improved performance of ACURATE *neo*2 THV. However, the durability of the device of up to 10 years will be investigated in the ongoing, randomized ACURATE–IDE trial (NCT03735667).

## 5. Conclusions

In conclusion, quantitative aortography shows a lower rate of moderate or severe paravalvular AR in what is the first European experience of the new-generation, self-expanding ACURATE *neo*2 THV when compared to the first-generation ACURATE *neo* THV. Further investigation is needed to confirm this finding. In addition, aortographic data need to be correlated and compared to Core Laboratory-adjudicated 30-day echocardiographic data.

## Figures and Tables

**Figure 1 jcm-10-04627-f001:**
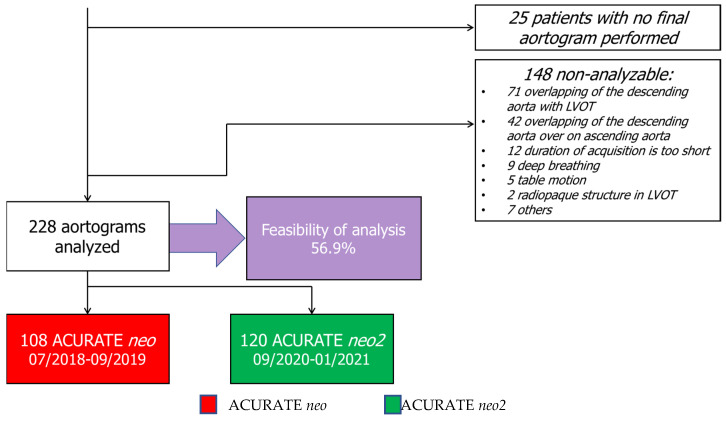
A flowchart of Core Laboratory quantitative assessment of AR.

**Figure 2 jcm-10-04627-f002:**
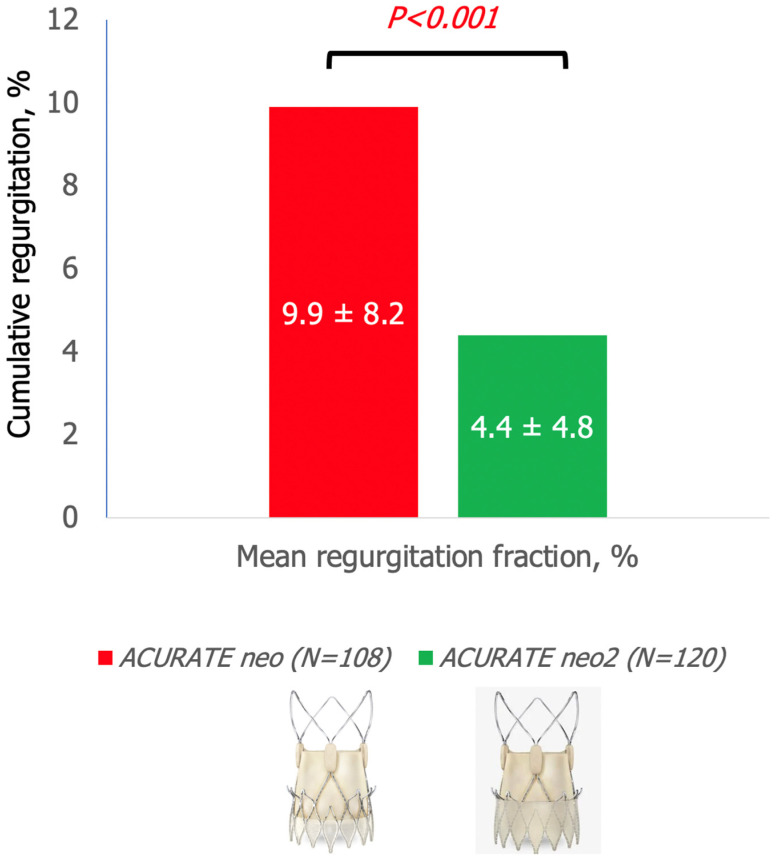
Comparison of mean regurgitation fraction on qAR following TAVI between ACURATE *neo* and ACURATE *neo*2 THVs. Mean aortic regurgitation fraction following TAVI obtained by qAR (4.4 ± 4.8% vs. 9.9% ± 8.2%; *p* < 0.001) was lower in the ACURATE *neo*2 THV cohort when compared with the ACURATE *neo* THV cohort.

**Figure 3 jcm-10-04627-f003:**
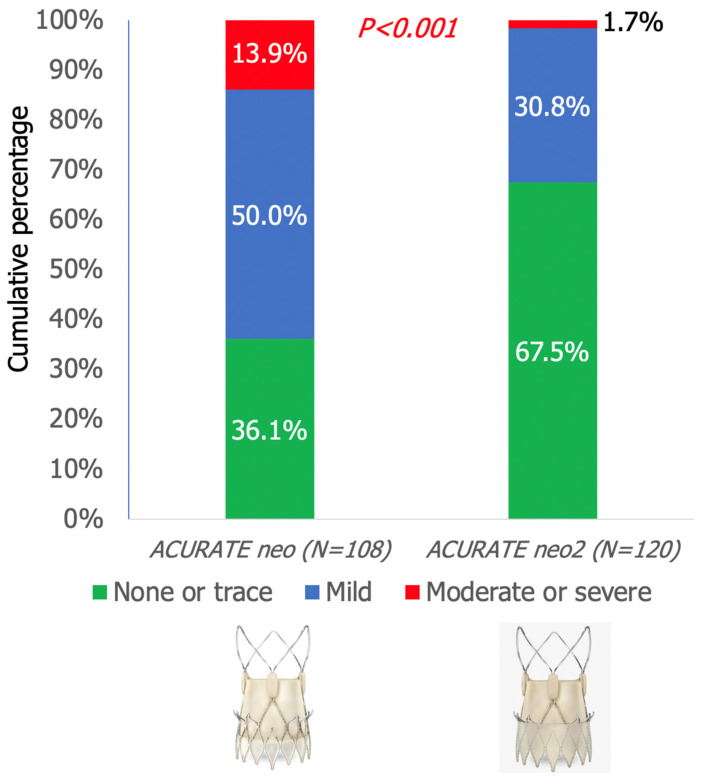
Cumulative percentage of AR severity grade on qAR following TAVI for ACURATE *neo* and ACURATE *neo*2 THVs. Moderate or severe qAR was seen in 1.7% vs. 13.9% (*p* < 0.001) in the ACURATE *neo*2 THV cohort when compared to the ACURATE *neo* THV cohort, respectively.

**Figure 4 jcm-10-04627-f004:**
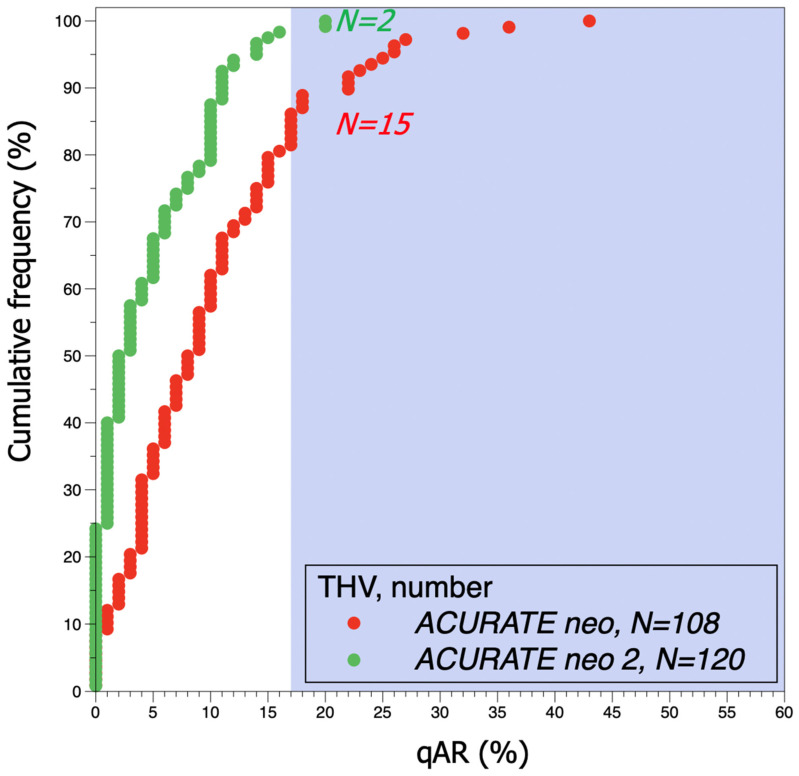
Cumulative frequency curves of qAR after TAVI for ACURATE *neo* and ACURATE *neo*2 THVs. The shaded background shows the area above 17% of qAR, indicating moderate or severe AR. Moderate or severe qAR was seen in 2 vs. 15 patients (*p* < 0.001) in the ACURATE *neo*2 THV cohort when compared to the ACURATE *neo* THV cohort, respectively. qAR: quantitative angiographic aortic regurgitation; AR: aortic regurgitation; THV: transcatheter heart valve; TAVI: transcatheter aortic valve replacement; LVOT: left ventricular outflow tract.

**Table 1 jcm-10-04627-t001:** The baseline and procedural characteristics between qAR-analyzable patients after ACURATE *neo*2 and ACURATE *neo* implantation.

	ACURATE *neo*2 *N* = 120	ACURATE *neo* *N* = 108	*p*-Value
**Baseline characteristics**			
Age	80.9 ± 6.1	80.4 ± 6.2	0.485
Man	43 (35.8)	51 (47.2)	0.081
Body weight, kg	72.3 ± 14.1	70.9 ± 13.2	0.440
Body height, cm	167.3 ± 9.1	167.0 ± 9.4	0.791
Hypertension	96 (80.0)	82 (75.9)	0.458
Diabetes mellitus	41 (34.2)	30 (27.8)	0.298
Atrial fibrillation	48 (40.0)	39 (36.1)	0.546
Prior stroke	14 (11.7)	10 (9.3)	0.554
Prior pacemaker implantation	13 (10.8)	16 (14.8)	0.368
Prior cardiac surgery	19 (15.8)	14 (13.0)	0.539
Previous percutaneous coronary intervention	28 (23.3)	33 (30.6)	0.219
Chronic obstructive pulmonary disease	21 (17.5)	18 (16.7)	0.867
NYHA 3 or 4	88 (73.3)	68 (63.0)	0.093
Creatinine clearance, mL/min	92.0 ± 33.2	85.1 ± 23.7	0.078
Euro score II, %	4.8 ± 3.7	5.5 ± 6.7	0.374
**Baseline Echocardiographic Parameters**			
Left ventricular ejection fraction <50%	16 (13.3)	24 (22.2)	0.078
LV Aorta mean gradient, mmHg	44.3 ± 15.3	47.1 ± 12.6	0.140
Systolic pulmonary artery pressure, mmHg	29.4 ± 23.4	30.5 ± 20.7	0.717
Aortic regurgitation before TAVI			0.511
None or trace	33 (47.8)	36 (52.2)	
Mild	72 (60.0)	63 (58.9)	
Moderate	12 (10.0)	7 (6.5)	
Severe	3 (2.5)	1 (0.9)	
Mitral regurgitation before TAVI			0.508
None or trace	16 (13.3)	21 (19.8)	
Mild	86 (71.7)	70 (66.0)	
Moderate	17 (14.2)	13 (12.3)	
Severe	1 (0.8)	2 (1.9)	
**Baseline Computed Tomography Findings**			
Perimeter derived mean annulus diameter, mm	23.6 ± 1.8	23.9 ± 1.7	0.124
Bicuspid aortic valve	12 (10.0)	11 (10.2)	0.963
**Procedural Characteristics**			
Predilatation	84 (70.0)	108 (100)	<0.001
Predilatation balloon size, mm	22.6 ± 1.7	22.2 ± 1.6	0.114
Postdilatation	52 (43.3)	60 (55.6)	0.065
Postdilatation balloon size, mm	23.0 ± 1.7	23.0 ± 1.7	0.952
Implanted THV size, mm	25.2 ± 1.6	25.6 ± 1.5	0.054
**Complications Following TAVI**			
Valve embolization	3 (2.5)	1 (0.9)	0.366
Need for second TAVI valve	2 (1.7)	0	0.276
Cardiac tamponade	1 (0.8)	0	0.526
New permanent pacemaker implantation	5 (4.2)	8 (7.4)	0.292
Major vascular complications	2 (1.7)	0	0.276
Major bleeding	2 (1.7)	1 (0.9)	0.624
Stroke	4 (3.3)	0	0.075
Mortality up to 30 days	0	0	-

NYHA: New York Heart Association; TAVI: transcatheter aortic valve replacement; LV: left ventricular; THV: transcatheter heart valve.

## Data Availability

Data is contained within the article.

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
