# Peer review of "Paravalvular Aortic Regurgitation Severity Assessed by Quantitative Aortography: ACURATE neo2 versus ACURATE neo Transcatheter Aortic Valve Implantation"

_jcm, 2021, doi:10.3390/jcm10204627_

Round 1

Reviewer 1 Report

The manuscript presents a comparison between two different generations of AcurateNeo THV, with the significant finding of a reduction in paravalvular leak incidence with the new AcurateNEO2 valve, as assessed by a videodensitometric angiographic assessment modality. The setting and results are clearly presented, and the overall quality of the manuscript is high. 

The main question addressed by the authors is relevant because it deals with a limit of first generation Acurate prostheses, the higher rate of residual aortic regurgitation in comparison with other THV platforms, that is related with worse outcomes on follow up. The paper shows a significant reduction in post-implant AR, although it doesn’t provide information on clinical outcomes (30 days or longer term survival or hospitalizations).

The topic is interesting: if the reduction in post-implant AR will be confirmed and a correlation with an improvement in clinical outcomes will be demonstrated it could lead to a wider use of this THV platform. The conclusions are consistent with the results presented and do address the main question.

The paper is well written, the text is clear and easy to read.

Author Response

We would like to thank this reviewer for encouraging and positive comments regarding our manuscript.

Reviewer 2 Report

Colleagues Ruck, Kim, Kawashima et al. report in their submitted manuscript "Paravalvular Aortic Regurgitation Severity Assessed by Quan- 2
titative Aortography: ACURATE neo2 versus ACURATE neo 3
Transcatheter Aortic Valve Implantation" a significant reducation of paravalvular aortic regurgitation of the ACURATE neo2 in comparison to the ACURATE neo valve, assessed by quantitative videodensitrometic angiography. The mansucript is well written and provides very interesting data on qAR and valve development. Although there are some comments, i would like to adress.

Major:

Correlation to post-TAVR echocardiographic is a major limitation of this study, as this technique is still state-of-the art. Is there no echocardiographic data available? This could help to differentiate the grading of AR in patients, where qAR was not feasible.  Some echocardiographic data should be available, as you mention in the discussion ("the next logic step is to correlate and compare aortographic data to Core Laboratory adjudicated 30-day echocardiographic data") and would improve the value of the manuscript significantly. 

Another major limitation is the low rate of analyzable angiography of 56.9%, authors should emphasize this limitiation, especially in respect to the use of the method in daily clinicial practice. Please name reasons for low feasibility of qAR in the text.

Were the investigators blinded? difference of acurate neo2 vs acurate neo in angiography detectable? Please comment in the methods section or in the limitiations.

What were the reasons for 2nd valves in the ACURATE neo2 cohort?

How were valve embolizations detected? Please comment the valve embolization rate in the ACURATE neo2 cohort. New valve design might reduce AR but increase the risk of embolisation.

Minor:

Feasibility of analysis: Numbers in text and Figure 1 should be the same (56.9% vs. 60.6%)

Lines 171/172: Wrong comma: "...is associated with, a significant reduction in..."

Figures: Please use consistent labels, font, font-size etc.

Author Response

We are grateful for the opportunity to respond to the major issues raised by the reviewers. We believe that the manuscript now has improved after the changes made per the reviewer’s suggestions. All changes in text are highlighted in red.

Reviewer 2

Colleagues Ruck, Kim, Kawashima et al. report in their submitted manuscript "Paravalvular Aortic Regurgitation Severity Assessed by Quantitative Aortography: ACURATE neo2 versus ACURATE neo Transcatheter Aortic Valve Implantation" a significant reduction of paravalvular aortic regurgitation of the ACURATE neo2 in comparison to the ACURATE neo valve, assessed by quantitative videodensitrometic angiography. The manuscript is well written and provides very interesting data on qAR and valve development. Although there are some comments, i would like to address.

Major comments 1:

Correlation to post-TAVR echocardiographic is a major limitation of this study, as this technique is still state-of-the art. Is there no echocardiographic data available? This could help to differentiate the grading of AR in patients, where qAR was not feasible.  Some echocardiographic data should be available, as you mention in the discussion ("the next logic step is to correlate and compare aortographic data to Core Laboratory adjudicated 30-day echocardiographic data") and would improve the value of the manuscript significantly. 

Answer to major comments 1

We thank this reviewer for his/her comments.

We do not have the echocardiographic data in the ACURATE neo cohort. The multicenter Early Neo2 Registry (NCT04810195) will correlate and compare aortographic data to Core Laboratory adjudicated 30-day echocardiographic data. Therefore, in this manuscript, we did not show the echocardiographic data both in the ACURATE neo2 and ACURATE neo cohorts.

We partially agree with the reviewer that quantitative angiographic videodensitometric aortic regurgitation (qAR) is not yet standard of care. However, the qAR is nowadays considered as an objective, accurate, and reproducible tool to assess aortic regurgitation (AR) after transcatheter aortic valve implantation (TAVI) [1-13]. This qAR by the CAAS A-Valve software (Pie Medical, Maastricht, The Netherlands) has been extensively vetted and validated in-vitro [14] [15] and in-vivo [16]. In the clinical setting, previous studies have demonstrated that qAR >17% corresponded to moderate or severe AR in comparison with trans-thoracic or trans-esophageal echocardiography, and was associated with an increase in mortality after TAVI [8],[12]. Furthermore, it has been shown to have an analysability yield greater than 95% in patients as determined in the ASSESS-REGURGE study [7]. To date, our core lab (CORRIB Research Center for Advanced Imaging and Core Lab, Galway, Ireland) database with 11 different types of transcatheter heart valve (THV) including the Myval, Venus A, VitaFlow, and ACURATE neo2 THVs pools more than 3,000 cases [5]. In addition, the randomized LANDMARK trial comparing the Myval, Evolut series and Sapien 3 series includes the qAR assessment as part of the primary composite end point in the study protocol [3]. In addition, in the Early Neo2 Registry on the ACURATE neo2, qAR has been selected by Boston Scientific as the minimalist method to assess AR in 2,000 patients.

In the VARC-3 criteria recently published [17],  AR by videodensitometry is acknowledged as a valid quantitative assessment, although Doppler echocardiography remains by tradition and convention the primary modality for assessing and comparing regurgitation after TAVI. Therefore, we edited the abstract to emphasize the echocardiographic results as current standard of care in the assessment of AR following TAVI.

Major comments 2

Another major limitation is the low rate of analyzable angiography of 56.9%, authors should emphasize this limitation, especially in respect to the use of the method in daily clinical practice. Please name reasons for low feasibility of qAR in the text.

Answer to major comments 2

As suggested, we named the reasons for the cases assessed as non-analyzable in the Result section as follows:

The common causes of non-analyzability of post-TAVI aortograms were listed in Figure 1. Out of 148 non-analyzable cases, the major reasons assessed as non-analyzable were overlapping of the descending aorta with LVOT (48.0%) or overlapping of the descending aorta on ascending aorta (28.4%) (Figure 1).

As stated before, the causes of non-analyzability in retrospective series have been extremely vetted in several publications with the overlap between the contrast in the descending aorta and contrast in either the reference region in the aortic root or the region of interest in LVOT as the main reason.  In both the ASSESS-REGURGE [7] and the OVAL [1] studies, the analyzability in prospective acquisition reached more than 90%.

Major comments 3

Were the investigators blinded? difference of acurate neo2 vs acurate neo in angiography detectable? Please comment in the methods section or in the limitations.

Answer to major comments 3

Aortographic data were analyzed in an independent Core Laboratory (CORRIB Research Center for Advanced Imaging and Core Lab, Galway, Ireland) by experienced analysts who were blinded to the investigators and other clinical data. When analyzed the angiographies, the difference of ACURATE neo2 and ACURATE neo THVs was not detectable. The following sentence was added to the Method section in the revised manuscript:

Aortographic data were analyzed in an independent Core Laboratory (CORRIB Research Center for Advanced Imaging and Core Lab, Galway, Ireland) by experienced analysts who were blinded to the investigators and other clinical data. When analyzed the angiographies, the difference of ACURATE neo2 and ACURATE neo THVs was not detectable.

Major comments 4

What were the reasons for 2nd valves in the ACURATE neo2 cohort?

How were valve embolization detected? Please comment the valve embolization rate in the ACURATE neo2 cohort. New valve design might reduce AR but increase the risk of embolization.

Answer to major comments 4

In the ACURATE neo2 cohort, there were 2 cases of second TAVI valve implantation which was due to valve embolization during retrieval of the delivery system. The embolized valves were brought to the aortic arch, and second valves were implanted successfully. The ACCURATE neo2 platform must not increase the risk of valve embolization since It's the same design as the ACCURATE neo except for the sealing skirt.

Minor comments

Feasibility of analysis: Numbers in text and Figure 1 should be the same (56.9% vs. 60.6%)

Lines 171/172: Wrong comma: "...is associated with, a significant reduction in..."

Figures: Please use consistent labels, font, font-size etc.

Answer to minor comments

We modified the text in the revised manuscript as follow:

Among 401 patients included in this study, 25 (6.2%) with no final aortogram performed, and qAR was analyzable in 228 (60.6%) patients including 120 and 108 patients treated with the ACURATE neo2 and ACURATE neo THVs, respectively.

Likewise, we improved the text and figures accordingly.

References

[1] Modolo R, van Mourik M, El Bouziani A, Kawashima H, Rosseel L, Abdelghani M, et al. Online Quantitative Aortographic Assessment of Aortic Regurgitation After TAVR: Results of the OVAL Study. JACC Cardiovasc Interv. 2021;14:531-8.

[2] Kawashima H, Wang R, Mylotte D, Jagielak D, De Marco F, Ielasi A, et al. Quantitative Angiographic Assessment of Aortic Regurgitation after Transcatheter Aortic Valve Implantation among Three Balloon-Expandable Valves. Glob Heart. 2021;16:20.

[3] Kawashima H, Soliman O, Wang R, Ono M, Hara H, Gao C, et al. Rationale and design of a randomized clinical trial comparing safety and efficacy of myval transcatheter heart valve versus contemporary transcatheter heart valves in patients with severe symptomatic aortic valve stenosis: The LANDMARK trial. Am Heart J. 2021;232:23-38.

[4] Modolo R, Chang CC, Onuma Y, Schultz C, Tateishi H, Abdelghani M, et al. Quantitative aortography assessment of aortic regurgitation. EuroIntervention. 2020;16:e738-e56.

[5] Modolo R, Chang CC, Abdelghani M, Kawashima H, Ono M, Tateishi H, et al. Quantitative Assessment of Acute Regurgitation Following TAVR: A Multicenter Pooled Analysis of 2,258 Valves. JACC Cardiovasc Interv. 2020;13:1303-11.

[6] Modolo R, Serruys PW, Chang CC, Wohrle J, Hildick-Smith D, Bleiziffer S, et al. Quantitative Assessment of Aortic Regurgitation After Transcatheter Aortic Valve Replacement With Videodensitometry in a Large, Real-World Study Population: Subanalysis of RESPOND and Echocardiogram Association. JACC Cardiovasc Interv. 2019;12:216-8.

[7] Modolo R, Chang CC, Tateishi H, Miyazaki Y, Pighi M, Abdelghani M, et al. Quantitative aortography for assessing aortic regurgitation after transcatheter aortic valve implantation: results of the multicentre ASSESS-REGURGE Registry. EuroIntervention. 2019;15:420-6.

[8] Tateishi H, Miyazaki Y, Okamura T, Abdelghani M, Modolo R, Wada Y, et al. Inter-Technique Consistency and Prognostic Value of Intra-Procedural Angiographic and Echocardiographic Assessment of Aortic Regurgitation After Transcatheter Aortic Valve Implantation. Circ J. 2018;82:2317-25.

[9] Miyazaki Y, Modolo R, Tateishi H, Serruys PW. Acute performance of first- and second-generation transcatheter aortic valves: a quantitative videodensitometric assessment of aortic regurgitation. EuroIntervention. 2018;14:e416-e7.

[10] Miyazaki Y, Modolo R, Abdelghani M, Tateishi H, Cavalcante R, Collet C, et al. The Role of Quantitative Aortographic Assessment of Aortic Regurgitation by Videodensitometry in the Guidance of Transcatheter Aortic Valve Implantation. Arq Bras Cardiol. 2018;111:193-202.

[11] Abdel-Wahab M, Abdelghani M, Miyazaki Y, Holy EW, Merten C, Zachow D, et al. A Novel Angiographic Quantification of Aortic Regurgitation After TAVR Provides an Accurate Estimation of Regurgitation Fraction Derived From Cardiac Magnetic Resonance Imaging. JACC Cardiovasc Interv. 2018;11:287-97.

[12] Abdelghani M, Tateishi H, Miyazaki Y, Cavalcante R, Soliman OII, Tijssen JG, et al. Angiographic assessment of aortic regurgitation by video-densitometry in the setting of TAVI: Echocardiographic and clinical correlates. Catheter Cardiovasc Interv. 2017;90:650-9.

[13] Tateishi H, Campos CM, Abdelghani M, Leite RS, Mangione JA, Bary L, et al. Video densitometric assessment of aortic regurgitation after transcatheter aortic valve implantation: results from the Brazilian TAVI registry. EuroIntervention. 2016;11:1409-18.

[14] Miyazaki Y, Abdelghani M, de Boer ES, Aben JP, van Sloun M, Suchecki T, et al. A novel synchronised diastolic injection method to reduce contrast volume during aortography for aortic regurgitation assessment: in vitro experiment of a transcatheter heart valve model. EuroIntervention. 2017;13:1288-95.

[15] Abdelghani M, Miyazaki Y, de Boer ES, Aben JP, van Sloun M, Suchecki T, et al. Videodensitometric quantification of paravalvular regurgitation of a transcatheter aortic valve: in vitro validation. EuroIntervention. 2018;13:1527-35.

[16] Modolo R, Miyazaki Y, Chang CC, Te Lintel Hekkert M, van Sloun M, Suchecki T, et al. Feasibility study of a synchronized diastolic injection with low contrast volume for proper quantitative assessment of aortic regurgitation in porcine models. Catheter Cardiovasc Interv. 2019;93:963-70.

[17] Varc-3 Writing C, Genereux P, Piazza N, Alu MC, Nazif T, Hahn RT, et al. Valve Academic Research Consortium 3: updated endpoint definitions for aortic valve clinical research. Eur Heart J. 2021;42:1825-57.

Reviewer 3 Report

Well written paper

Expected finding from a newer THV system but a nice addition to ever growing TAVR literature

In table1, only 21 patients had COPD in ACURATE Neo 2 group, so 53.8 as percentage is wrong.

Author Response

We would like to thank this reviewer for encouraging and positive comments regarding our manuscript. Accordingly, we modified table 1 in the revised manuscript.

Round 2

Reviewer 2 Report

The authors responded adequatly to the comments. No further comments.